# Effect of Reduction of Pt–Sn/α-Al₂O₃ on Catalytic Dehydrogenation of Mixed-Paraffin Feed

**Suresh Avithi Kanniappan [1,2]** and **Udaya Bhaskar Reddy Ragula [1,2,]\***

[1]  Department of Chemical Engineering and Materials Science, Amrita School of Engineering, Coimbatore, Amrita Vishwa Vidyapeetham, Coimbatore 641112, India; aksureshchem91@gmail.com

[2]  Centre of Excellence in Advanced Materials and Green Technologies, Amrita School of Engineering, Coimbatore, Amrita Vishwa Vidyapeetham, Coimbatore 641112, India

\*  Correspondence: u_bhaskarreddy@cb.amrita.edu

**Abstract:** The effect of the Pt–Sn/α-Al₂O₃ catalyst reduction method on dehydrogenation of mixed-light paraffins to olefins has been studied in this work. Pt–Sn/α-Al₂O₃ catalysts were prepared by two different methods: (a) liquid phase reduction with NaBH₄ and (b) gas phase reduction with hydrogen. The catalytic performance of these two catalysts for dehydrogenation of paraffins was compared. Also, the synergy between the catalyst reduction method and mixed-paraffin feed (against individual paraffin feed) was studied. The catalysts were examined using X-ray diffraction (XRD), scanning electron microscopy (SEM), energy dispersive X-ray spectroscopy (EDS), X-ray photoelectron spectroscopy (XPS), thermogravimetric analysis (TGA), and Brunauer–Emmett–Teller (BET) analysis. The individual and mixed-paraffin feed dehydrogenation experiments were carried out in a packed bed reactor fabricated from Inconel 600, operating at 600 °C and 10 psi pressure. The dehydrogenation products were analyzed using an online gas chromatograph (GC) with flame ionization detector (FID). The total paraffin conversion and olefin selectivity for individual paraffin feed (propane only and butane only) and mixed-paraffin feed were compared. The conversion of propane only feed was found to be 10.7% and 9.9%, with olefin selectivity of 499% and 490% for NaBH₄ and hydrogen reduced catalysts, respectively. The conversion of butane only feed was found to be 24.4% and 23.3%, with olefin selectivity of 405% and 418% for NaBH₄ and hydrogen reduced catalysts, respectively. The conversion of propane and butane during mixed-feed dehydrogenation was measured to be 21.4% and 30.6% for the NaBH₄ reduced catalyst, and 17.2%, 22.4% for the hydrogen reduced catalyst, respectively. The olefin selectivity was 422% and 415% for NaBH₄ and hydrogen reduced catalysts, respectively. The conversions of propane and butane for mixed-paraffin feed were found to be higher when compared with individual paraffin dehydrogenation. The thermogravimetric studies of used catalysts under oxygen atmosphere showed that the amount of coke deposited during mixed-paraffin feed is less compared with individual paraffin feed for both catalysts. The study showed NaBH₄ as a simple and promising alternative reduction method for the synthesis of Pt–Sn/Al₂O₃ catalyst for paraffin dehydrogenation. Further, the studies revealed that mixed-paraffin feed dehydrogenation gave higher conversions without significantly affecting olefin selectivity.

**Keywords:** dehydrogenation; Pt–Sn/Al₂O₃ catalyst; reduction method; mixed-paraffin feed

## 1. Introduction

In daily life, olefins like propylene, butylene, and butadiene are the main building blocks for the production of high demand chemicals and polymers such as propylene oxide, cumene, acrylonitrile, isopropyl alcohol, polypropylene, and polybutadiene [1,2]. The demand for olefins is expected to increase at a rate of 3.7% annually [3]. The primary routes for light olefins production are steam

cracking and dehydrogenation of individual paraffins (propane and butane) [2,4,5]. These individual paraffins are obtained from fluid catalytic cracking (FCC) unit, atmospheric, and vacuum distillation. The light olefins are also obtained as by-products in limited quantities from FCC unit along with light paraffins. Therefore, light paraffins are available as a mixture from the processes mentioned above, which require separation before they undergo dehydrogenation individually [6]. The products of individual paraffin dehydrogenation are to be separated again, owing to their wide range (refer to Figure 1a). The feed separation, dehydrogenation of individual paraffins, and product separation during individual paraffin dehydrogenation result in both high capital and operating costs. The total operating and capital cost can be minimized if the feed to the dehydrogenation reactor is directly taken from fluidized catalytic cracking unit. Therefore, the mixed-paraffin dehydrogenation process is preferable over individual paraffin dehydrogenation (refer to Figure 1b). Only a few works were reported on mixed feed dehydrogenation [7–9].

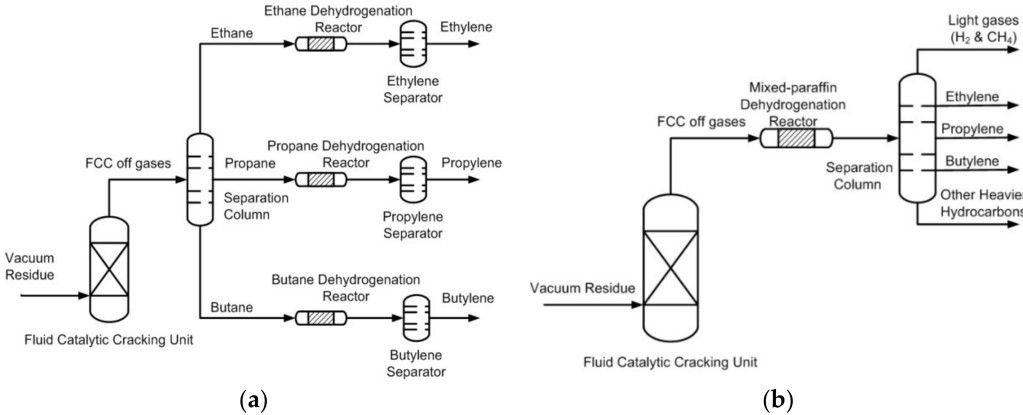

(**a**)　　　　　　　　　　　　　　　　(**b**)

**Figure 1.** Block diagram of the (**a**) current dehydrogenation process and (**b**) proposed dehydrogenation process. FCC, fluid catalytic cracking.

Paraffin dehydrogenation is a highly endothermic reaction that is limited by thermodynamic equilibrium. The individual paraffin dehydrogenation reactions along with the standard enthalpy of the reaction are given in Equations (1) and (2). These reactions are generally carried out at high temperatures (550–700 °C) and low pressure using different catalysts like noble metals (Pt, Cr, Ir, Rh, V, Pd) and their oxides [2,10,11]. The major types of catalyst deactivation during dehydrogenation are sintering and coking [10,11].

Dehydrogenation reactions:

$$C_3H_8 \rightarrow C_3H_6 + H_2 \quad \Delta H^o_{298} = 124.3 \, \text{kJ} \, \text{mol}^{-1}, \tag{1}$$

$$C_4H_{10} \rightarrow C_4H_8 + H_2 \quad \Delta H^o_{298} = 120.0 \, \text{kJ} \, \text{mol}^{-1}. \tag{2}$$

Hydrogenolysis reactions:

$$C_3H_8 + H_2 \rightarrow CH_4 + C_2H_6, \tag{3}$$

$$C_4H_{10} + H_2 \rightarrow 2CH_4 + C_2H_6. \tag{4}$$

Coke formation reaction:

$$R - CH_3 \rightarrow \text{Coke}. \tag{5}$$

The promoters like Sn, In, Ga, and Pb have been widely used to reduce the deactivation of the catalyst as a result of sintering of noble metal clusters and suppress the secondary reactions like hydrogenalysis (refer to Equations (3) and (4) and coking (Equation (5)) [7,11,12]. Among those, Pt promoted with Sn was reported to be the most efficient catalyst for paraffin dehydrogenation [13–16]. The Sn donates an electron to Pt, which reduces the adsorption of alkenes (products of dehydrogenation)



and increases the barrier for dissociative adsorption of light alkanes [17]. Considering the superior cleaving of the C–H bond compared with the C–C bond and stability, the Pt–Sn based catalysts were identified to be excellent catalysts for dehydrogenation [10,18]. $Al_2O_3$, $SiO_2$, SBA-15, and zeolites were widely used as catalyst supports for dehydrogenation [14,17,19–21]. Among the supports studied, $Al_2O_3$ has been reported to be a good support for the Pt–Sn catalyst owing to its high thermal stability and uniform pore size. Further, the nature of its limited acid sites reduces catalyst deactivation by coking [16,20].

In previous studies, the Pt–Sn/$Al_2O_3$ catalyst for paraffin dehydrogenation was studied in detail [20,22–28]. As chlorinated precursors were used for Pt–Sn metals, different methodologies were used for the removal of chlorine in impregnated catalysts to obtain metals and metal alloys. The elimination of chlorine had improved the stability and activity of the catalyst [22]. Zangeneh et al. [24] synthesized the Pt–Sn/$Al_2O_3$ catalyst with calcination in two stages at 350 °C and 550 °C, each for 2 h. Jung J.W. et al. [20] used hydrogen gas as a reducing agent without calcination, at different reduction temperatures, and reported 550 °C as the optimum temperature for direct reduction. Zhang H. et al. [22] used an air–steam mixture as a reducing agent at 540 °C. Li Q. et al. [25] used calcination at 530 °C followed by steam treatment. Zhang Y. et al. [26] synthesized Pt–Sn–Na/La-doped on $Al_2O_3$ support catalyst calcined at 500 °C for 4 h followed by hydrogen reduction at 500 °C for 8 h.

Mostly, hydrogen was used as a reducing agent at >550 °C for catalyst reduction [20,21,23,24]. Considering the large-scale synthesis of catalysts, cost, and demand for hydrogen gas, finding an alternative reducing agent minimizes the catalyst production cost and improves safety while synthesizing the catalysts for dehydrogenation. The liquid phase sodium borohydride ($NaBH_4$) is one of the powerful reducing agents, which is widely used to reduce aldehydes, and ketene and is a potential alternative to gas phase hydrogen. There are very few studies reported on $NaBH_4$ as a reducing agent for Pt–Sn catalysts [29,30].

The objectives of this study are (a) to study the effect of reduction methodology on dispersion of the catalysts, (b) to compare the performance of hydrogen reduced catalyst (in the gas phase) with that of NaBH4 reduced catalyst (in the liquid phase) for paraffin dehydrogenation, and (c) to compare individual paraffin feed with mixed-paraffin feed on the extent of paraffin dehydrogenation and olefin selectivity. The extent of dehydrogenation was measured in terms of paraffin conversion and olefin selectivity.

## 2. Results and Discussion

### 2.1. Characterization Results

#### 2.1.1. Brunauer–Emmett–Teller (BET) Measurements

BET analysis was used to measure the surface area, average pore diameter, and pore volume of $NaBH_4$ reduced and $H_2$ reduced catalysts. The results of the BET analysis are provided in Table 1. The error in the surface area measurements as specified by the instrument supplier is a maximum of 5%. The $H_2$ reduced catalyst shows high surface area, average pore diameter, and pore volume. The low surface area of $NaBH_4$ reduced catalyst might be because of the reduction after calcination during the catalyst synthesis process. The pore diameter for both catalysts after reduction is similar, that is, approximately 2 nm. Comparing the range of the diameter of the catalyst, this will be in micropore regime. A similar range of pore diameter for the Pt–Sn/$\alpha$-$Al_2O_3$ catalyst was reported by Kogan S.B. et al. [31] and Ballarini A.D. et al. [32].

**Table 1.** Properties of NaBH$_4$ and hydrogen reduced catalysts.

| Catalyst Type | Surface Area [a] (m$^2$/g) | Avg. Pore Dia. [a] (nm) | Total Pore Vol. [a] (cm$^3$/g) | Pt [b] (wt%) | Sn [b] (wt%) |
|---|---|---|---|---|---|
| NaBH$_4$ reduced | 6 | 2.75 | 0.082 | 6.61 | 2.36 |
| H$_2$ reduced | 15 | 2.02 | 0.150 | 2.30 | 0.56 |

[a] By nitrogen adsorption–desorption studies (Brunauer–Emmett–Teller (BET)). [b] By energy dispersive X-ray spectroscopy (EDX).

### 2.1.2. XRD Analysis

The XRD analysis was performed on Pt–Sn/α-Al$_2$O$_3$ catalysts reduced with NaBH$_4$ and hydrogen, and the comparison for the same is shown in Figure 2a. The characteristic peaks for both the catalysts were obtained in the same 2θ value. For PtSn alloys, the characteristic peaks must be between 36° and 48° in 2θ value [30,33]. Peaks at 39° and 45.5° correspond to Pt$_3$Sn alloy phase, whereas peaks at 41.7° and 44° in 2θ value correspond to the PtSn alloy phase [14,20]. Further, it was also reported that the Pt$_3$Sn alloy phase was only formed at Pt/Sn molar ratios greater than 1 [14]. In the current experimental studies, the nominal Pt/Sn molar ratio was 0.4. This means that there are greater chances for the formation of both PtSn and Pt$_3$Sn alloys.

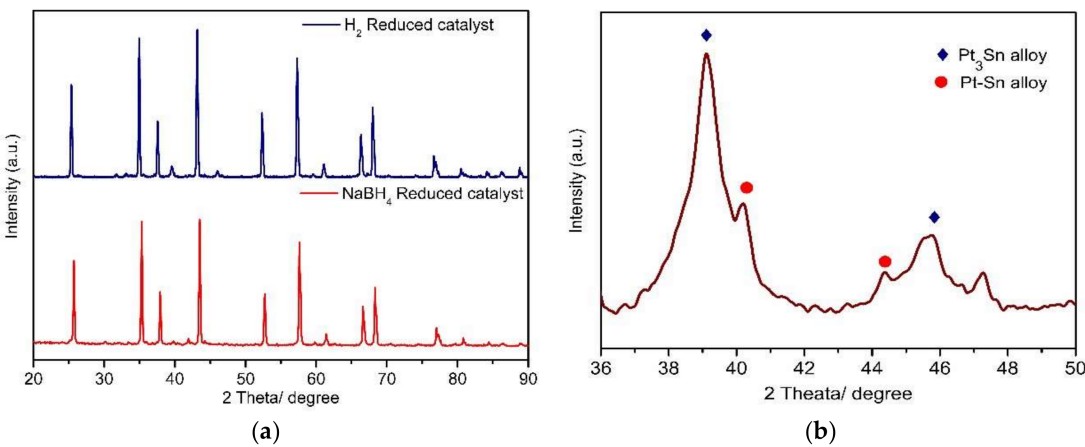

(a)    (b)

**Figure 2.** X-ray diffraction (XRD) patterns of (**a**) Pt–Sn/Al$_2$O$_3$ NaBH$_4$ and H$_2$ reduced catalyst and (**b**) Pt–Sn precursor without alumina support.

Figure 2a for the hydrogen reduced catalyst shows lower intensity peaks at 39° and 45.5° and 41.7° and 44°. These lower intensities of the characteristic peaks for PtSn alloys were overshadowed by the peaks of α-Al$_2$O$_3$ support, which is because of lower amounts of Pt and Sn and the highly crystalline nature of α-Al$_2$O$_3$. The obtained peaks from XRD analysis were identified using Joint Committee on Powder Diffraction Standards (JCPDS) file no: 71-1123. The high intensity peaks with 2θ value of 25.59°, 35.14°, 37.48°, 43.36°, 52.55°, and 57.52° are assigned to (0 1 2), (1 0 4), (1 1 0), (1 1 3), (2 0 2), (0 2 4), and (1 1 6) planes of hexagonal phase α-Al$_2$O$_3$ respectively [16]. For NaBH$_4$ reduced catalysts, the peaks were not observed, which can be seen in Figure 2a. This might be because of the finely dispersed nature of PtSn alloys on the support or because it is too small for the XRD detection limit. The finely dispersed nature of the PtSn alloys is evident from the SEM images of the catalysts for NaBH$_4$ reduced catalyst (refer to Figure 3a for the NaBH$_4$ reduced catalyst and Figure 3b for the hydrogen reduced catalyst in Section 2.1.3), Similar results were reported by Vaidya et al. [30] and Lee J. K. et al. [33].

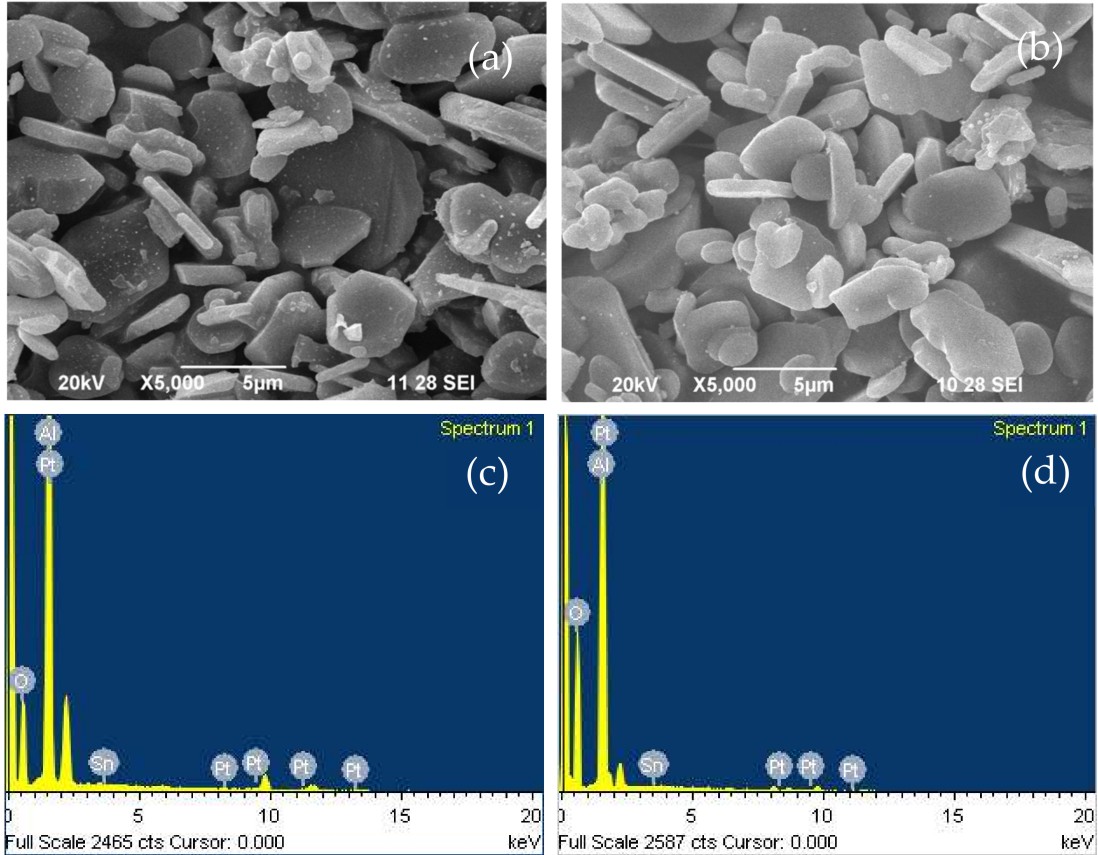

**Figure 3.** Scanning electron microscopy (SEM) morphology of Pt–Sn/Al$_2$O$_3$ reduced with (**a**) NaBH$_4$ and (**b**) H$_2$, and energy dispersive X-ray spectroscopy (EDS) mapping of Pt–Sn/Al$_2$O$_3$ reduced with (**c**) NaBH$_4$ and (**d**) H$_2$.

To suppress the Al$_2$O$_3$ crystalline effect for XRD measurements, the mixer of Pt and Sn precursor solutions was directly dried without α-Al$_2$O$_3$ support at 120 °C for 12 h followed by calcination at 550 °C for 6 h to confirm the formation of PtSn alloys. Figure 2b shows the XRD peaks for the unsupported PtSn sample. The distinct peaks associated with Pt$_3$Sn alloy at 39° and 45.4° (JCPDS 00-035-1360) and PtSn alloy at 41.7° and 44° (JCPDS 00-025-0614) were found. These results also match the existing data available in the literature reported by Kaylor M. et al. [17] and Jung J.W. et al. [20].

From Figure 2a,b, it is confirmed that both PtSn and Pt$_3$Sn alloys on Al$_2$O$_3$ support. Further, by comparing the XRD analysis results with SEM results, it is confirmed that the PtSn alloys are finely dispersed on Al$_2$O$_3$ support for the NaBH$_4$ catalyst when compared with the hydrogen reduced catalyst, that is, the particle size of PtSn alloys is larger on the hydrogen reduced catalyst when compared with the NaBH$_4$ reduced catalyst. The effect of dispersion of PtSn alloy phases on the performance during dehydrogenation experiments and the amount of coke deposited is discussed in Section 2.2.

### 2.1.3. SEM and EDX Analysis

SEM analysis was performed on Pt–Sn/α-Al$_2$O$_3$ reduced with NaBH$_4$ and H$_2$, to understand the morphological structure of catalyst surface and the distribution of active components in the support. The bright dotted clusters show the presence and distribution of Pt and Sn metallic species on Al$_2$O$_3$ surface. NaBH$_4$ reduced catalyst shows higher distribution of Pt and Sn, shown in Figure 3a, compared with the H$_2$ reduced catalyst in Figure 3b. The elemental mapping was measured using EDX analysis, shown in Figure 3c,d, and the metallic weight% is presented in Table 1. As EDS is a local or point characterization method, for finding the metal mapping on a surface, the presented metal% may vary

because of the non-uniform dispersion of the active metals on the support. The EDS result confirms the uniform dispersion of active metals (or bimetallic Pt and Sn alloys) over alumina support.

### 2.1.4. XPS Analysis

XPS analysis was performed to measure the metallic state of Pt and Sn in $NaBH_4$ reduced and hydrogen reduced catalysts. The XPS spectra were deconvoluted using CasaXPS software. The XPS analysis of both catalysts is shown in Figure 4. The solid colored lines (except black color) show the deconvoluted curves. The spectrum of Sn 3d and Pt 4f core levels are shown in Figure 4a,b, respectively. The binding energies of zero valent Sn $3d_{5/2}$ and oxidized Sn(II.IV) were obtained at 485 eV and 486.5 eV, respectively, as shown in Figure 4a. The zero valent Sn peak was found in the hydrogen reduced catalyst, whereas both Sn(0) and Sn(II,IV) peak lines were seen in the $NaBH_4$ reduced catalyst [13,16,30].

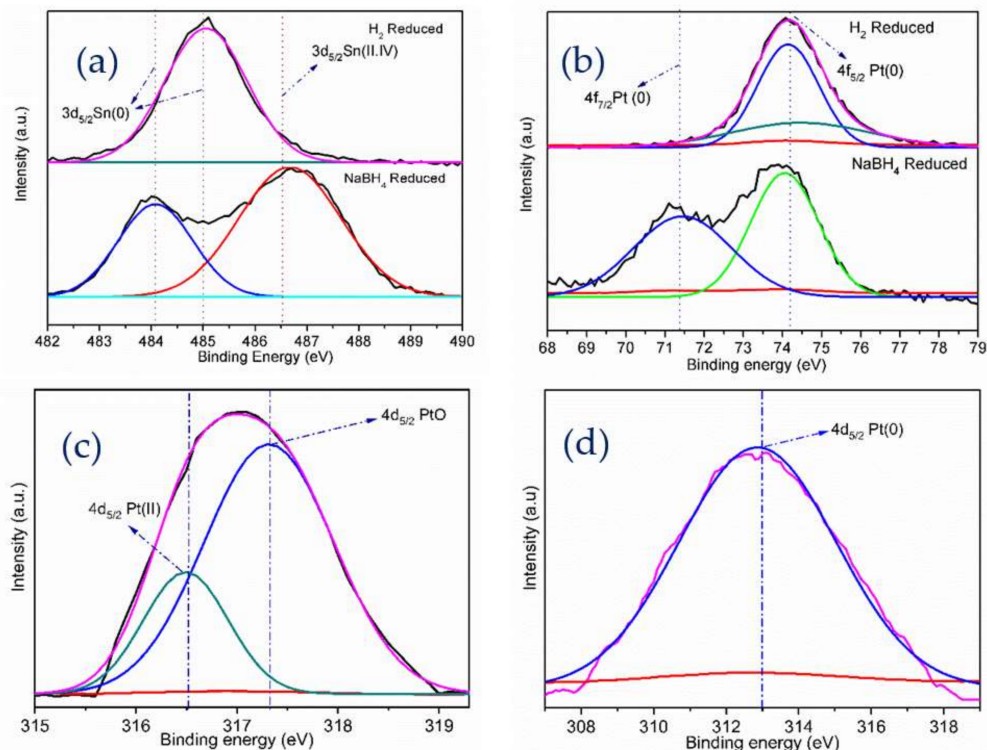

**Figure 4.** XPS spectra for the Pt–Sn–$Al_2O_3$ catalyst (**a**) Sn 3d, (**b**) Pt 4f, and (**c**) Pt 4d of the $NaBH_4$ reduced catalyst; and (**d**) Pt 4d of the $H_2$ reduced catalyst.

Figure 4b shows the zero valent Pt at $4f_{7/2}$ and $4f_{5/2}$ peaks at 71.3 eV and 74.5 eV [34]. However, this energy region was overlapped by the Al 2p peak of supporting material. Therefore, the Pt 4d lines analyzed for both the hydrogen reduced and $NaBH_4$ reduced catalyst are shown in Figure 4c,d. Pt(II) and PtO were found at 316.5 eV and 317.3 eV for the $NaBH_4$ reduced catalyst [30,35], whereas Pt species were completely reduced to zero valent platinum in hydrogen reduced catalyst at a binding energy of 313 eV [36]. A slight shift in binding energies for Pt(O) and Pt(II) peaks was observed, compared with monometallic Pt. This shift might be because of a small variation in the electronic configuration of Pt atoms on the surface. This confirms the formation of the Pt–O–Sn interaction. Similar results were reported by Vaidya S.H. et al. [30]. Generally, the metallic state of Sn acts as a poisoning agent and Sn(II.IV) serves as promotor in the catalyst [1,33].

The comparison of metallic Pt $4f_{5/2}$ peak areas between hydrogen and $NaBH_4$ reduced catalysts reveals that the Pt was less distributed in hydrogen reduced catalyst, and metallic Pt $4f_{7/2}$ was not seen in the hydrogen reduced catalyst, which is concurrent with the XRD and SEM results on the dispersion

of the catalysts. In metallic Sn $3d_{5/2}$, the areas for both catalysts are almost same, indicating uniform distribution of Sn.

### 2.2. Dehydrogenation Catalytic Performance Studies

#### 2.2.1. Blank Test

The reactions mentioned in Equations (1) to (5) are general reactions under dehydrogenation conditions. Apart from these reactions, thermal cracking reaction of propane to give ethylene and methane and thermal cracking of butane to produce ethane and ethylene may also occur depending on the system [37]. These reactions also occur in the absence of catalyst. The black dehydrogenation experiments were carried out to measure the extent of the thermal cracking reaction at the reactor and flow conditions. The blank test was conducted in an empty reactor (without catalyst) at the same experimental conditions discussed in Section 3.3 (Dehydrogenation of Individual and Mixed Paraffin Feed). From the results obtained from the product analysis using gas chromatograph (GC), it was found that the conversion due to thermal cracking was negligible for the chosen experimental conditions.

#### 2.2.2. Individual Paraffin Dehydrogenation

Propane and butane (individual feeds) were individually dehydrogenated over NaBH$_4$ and hydrogen reduced Pt–Sn/Al$_2$O$_3$ catalysts. The comparisons of performance of these two catalysts are given in Figure 5a (propane feed only) and Figure 5b (butane feed only). The propane and butane conversions are calculated using Equations (8) and (9), and the olefin selectivity is calculated using Equation (10). It should be noted that only relevant terms that are applicable for each experiment in Equation (10) were used for calculations. The conversion and selectivity are significantly different for both individual feeds (i.e., propane alone and butane alone) for a reaction period of 240 min. This might be because of the kinetic parameters that are associated with the reactions. Figure 5a gives the comparison of paraffin (propane) conversion and olefin selectivity for both catalysts for propane feed only. For the hydrogen reduced catalyst, the initial and final propane conversions were found to be 18.3% and 9.9%, respectively. Meanwhile, the initial and final propane conversion for the NaBH$_4$ catalyst were found to be 12.1% and 10.7%, respectively. This suggests that the hydrogen reduced catalyst had undergone a faster deactivation when compared with the NaBH$_4$ reduced catalyst, which is evident from the results shown in Figure 5a for propane only feed. The initial and final olefin selectivity was calculated to be 497% and 490%, respectively, for the hydrogen reduced catalyst. The initial and final olefin selectivity was calculated to be 522% and 499%, respectively, for the NaBH$_4$ reduced catalyst. Table 2 shows the comparison of final paraffin conversion and olefin selectivity (after 240 min) for both catalysts for individual paraffin dehydrogenation experiments.

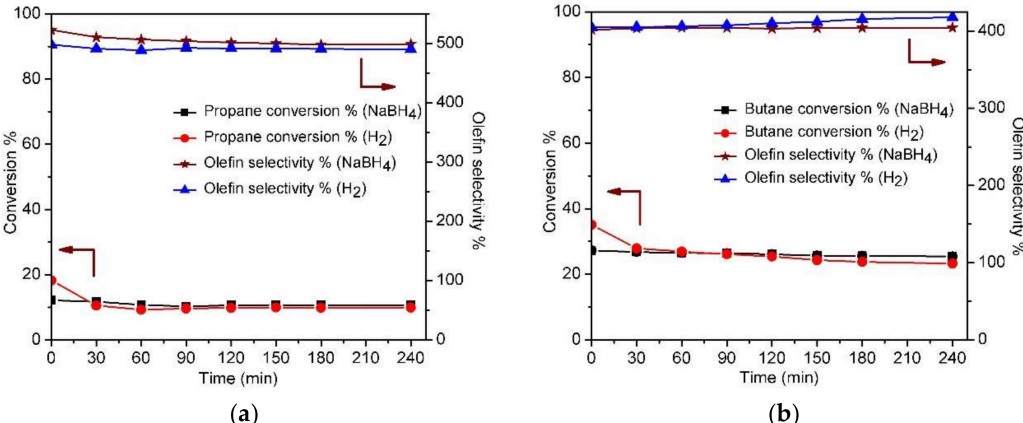

**Figure 5.** Conversion and selectivity over NaBH$_4$ and H$_2$ reduced catalysts on (**a**) propane only feed and (**b**) butane only feed.

**Table 2.** Conversion and selectivity for individual paraffin feed dehydrogenation after 240 min.

| Dehydrogeantion Parameter [a] | NaBH$_4$ Reduced Catalyst | | Hydrogen Reduced Catalyst | |
|---|---|---|---|---|
| | Propane Feed Only | Butane Feed Only | Propane Feed Only | Butane Feed Only |
| Propane conversion% | 10.7 | - | 9.9 | |
| Propylene selectivity% | 499 | - | 490 | |
| Butane conversion% | - | 25.4 | | 23.3 |
| Butylene selectivity% | - | 404 | | 418 |

[a] Process conditions: temperature—600 °C, reaction time—240 min, reactor pressure—10 psi, weight hourly space velocity (*WHSV*)—15 h$^{-1}$.

The catalyst may be deactivated as a result of sintering and coking. A deactivation parameter (D), given in Equation (6), is defined solely for this purpose to understand the extent of deactivation over 240 min. The deactivation parameters for propane only feed were calculated to be 11.6% and 45.8% for sodium borohydride reduced and hydrogen reduced catalysts, respectively. This indicates that the extent of deactivation was higher for the hydrogen reduced catalysts when compared with the sodium borohydride reduced catalyst. Similar ranges of conversion for propane dehydrogenation for were obtained for Pt–Sn/α-Al$_2$O$_3$ reduced with hydrogen [31].

$$D = \frac{X_o - X_f}{X_o} * 100, \qquad (6)$$

where

$X_o - Initial\ conversion,$
$X_f - Final\ conversion.$

Figure 5b shows the comparison of butane conversion and olefin selectivity for both catalysts for butane feed only dehydrogenation experiments. For the hydrogen reduced catalyst, the initial and final butane conversions were found to be 35.1% and 23.3%, respectively. Meanwhile, the initial and final butane conversions for the NaBH$_4$ catalyst were found to be 27.2% and 25.4%, respectively. The trend observed for butane conversions was similar to that of propane conversion for individual paraffin feed. This suggests that the hydrogen reduced catalyst had undergone a faster deactivation when compared with the NaBH$_4$ reduced catalyst. The initial and final olefin selectivity for hydrogen reduced catalyst was found to be 405% and 418%, respectively. Meanwhile, the initial and final olefin selectivity for the NaBH4 catalyst was found to be 401% and 404%, respectively. The final paraffin conversion and olefin selectivity (after 240 min) for both catalysts for individual paraffin dehydrogenation experiments are shown in Table 2. The deactivation parameter for the hydrogen reduced catalyst is 33.8%, and it is 6.7% for the NaBH$_4$ reduced catalyst. A similar range of results for butane conversion was obtained for the Pt–Sn/α-Al$_2$O$_3$ catalyst reduced with hydrogen [32].

For the individual paraffin feed, the initial conversion is generally higher for the hydrogen reduced catalyst when compared with the NaBH$_4$ reduced catalyst. However, the conversion for the hydrogen reduced catalyst declines rapidly. The final conversion (after 240 min) for NaBH$_4$ reduced catalysts is higher when compared with the hydrogen reduced catalyst. This indicates that the NaBH$_4$ reduced catalyst shows better stability and resistance to coking when compared with the hydrogen reduced catalyst.

As discussed in Section 2.1 from the XRD, SEM, and XPS results, the dispersion of PtSn alloys is lower on the hydrogen reduced catalyst when compared with the NaBH$_4$ reduced catalyst. This also indicates that the particle size of active alloys phase is larger in the case of the hydrogen reduced catalyst. The faster deactivation of hydrogen reduced catalyst when compared with the NaBH$_4$ reduced catalyst as a result of coke deposit on the larger particle size of PtSn alloys on the support. The comparison of the amount of coke deposition under different scenarios is discussed in Section 2.2.4 though thermogravimetric analysis (TG)-differential thermal analysis (DTA) under oxygen atmosphere.

### 2.2.3. Mixed Paraffin Dehydrogenation

The mixed-paraffin feed dehydrogenation experiments were performed using a mixture of propane and butane gases using both $NaBH_4$ and hydrogen reduced catalysts. The results of the mixed-paraffin feed dehydrogenation are shown in Figure 6. The initial and final conversion of propane under mixed feed condition was found to be 23.2% and 17.2%, respectively, for the hydrogen reduced catalyst. The initial and final conversion of propane under mixed feed condition was found to be 24% and 21.4%, respectively, for the $NaBH_4$ reduced catalyst. It is interesting to note that the initial propane conversion was the same under mixed-paraffin feed conditions for both catalysts and the final conversion for $NaBH_4$ reduced catalyst is higher than that of the hydrogen reduced catalyst. Table 3 gives the final (after 240 min) paraffin conversion and olefin selectivity for both catalysts.

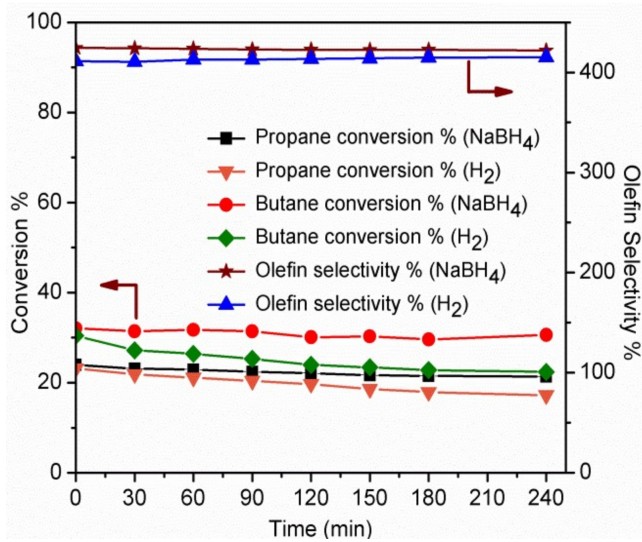

**Figure 6.** Conversion and Selectivity over the $NaBH_4$ and $H_2$ reduced catalysts for mixed-paraffin feed.

**Table 3.** Conversion and selectivity mixed paraffin feed dehydrogenation after 240 min.

| Dehydrogenation Parameter [a] | NaBH₄ Reduced Catalyst | Hydrogen Reduced Catalyst |
|---|---|---|
| Propane conversion% | 21.4 | 17.2 |
| Butane conversion% | 30.6 | 22.4 |
| Total Olefin Selectivity% | 421.9 | 415.2 |

[a] Process conditions: temperature—600 °C, reaction time—240 min, reactor pressure—10 psi, *WHSV*—15 h$^{-1}$.

The initial and final conversion of butane under mixed feed condition was found to be 30.4% and 22.4%, respectively, for the hydrogen reduced catalyst. The initial and final conversion of butane under mixed feed condition was found to be 32.1% and 30.6%, respectively, for the $NaBH_4$ reduced catalyst. Though the initial conversion of butane for both the catalyst is the same under mixed paraffin feed conditions, it is interesting to note that the final conversion of butane for the $NaBH_4$ reduced catalyst is much higher, indicating lower deactivation of the $NaBH_4$ catalyst when compared with the hydrogen reduced catalyst.

Figure 6 also shows the total olefin selectivity for both catalysts under mixed feed conditions. The initial and final total olefin selectivity (calculated using Equation (10)) under mixed paraffin feed conditions was 411% and 415%, respectively, for the hydrogen reduced catalyst. Similarly, the initial and final total olefin selectivity was calculated to be 424% and 421%, respectively.

The higher conversions of propane and butane without affecting the olefin selectivity for $NaBH_4$ reduced catalyst are attributed to highly dispersed and fine catalyst particles, as discussed in the XRD analysis in Section 2.1.2. As shown in Figure 4a (XPS analysis—Section 2.1.4), the presence of Sn(II,IV)

of PtSn alloy phase in the $NaBH_4$ reduced catalyst reduces the catalyst deactivation due to coking. The faster deactivation of the hydrogen reduced catalyst is attributed to the presence of Sn(0).

The gas chromatograph (GC) analysis of fresh feed to the reactor for (A) propane only, (B) butane only, and (C) mixed feed is provided as a supplementary to the article. It is shown in Figure S1. The quantities of each species in the feed are also given in the supplementary information Table S1. Figure 7 shows the GC analysis of the product for (A) propane only, (B) butane only, and (C) mixed feed. The products during mixed-paraffin dehydrogenation are different from individual feed dehydrogenation. Cyclopropane and 1-pentene are additional products during butane only and mixed-feed dehydrogenation. Meanwhile, propadiene and acetylene were formed as additional products during propane only feed, which were not found during mixed-paraffin dehydrogenation. Similar results were reported by Pakhomov N. A. et al. [38] during oxidative dehydrogenation using ceria-alumina catalysts.

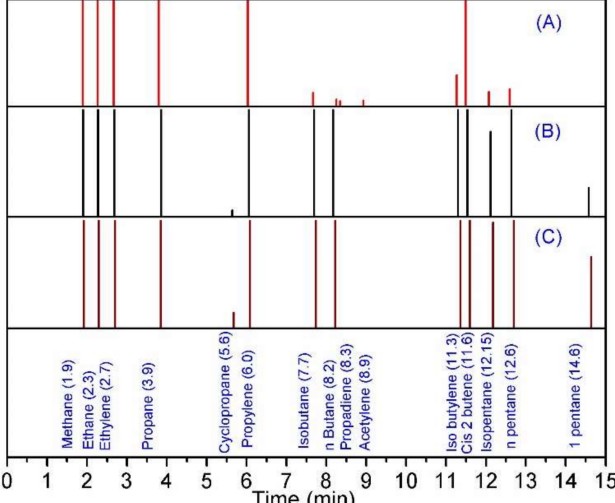

**Figure 7.** Product profiles obtained from gas chromatograph (GC) analysis: (**A**) propane only, (**B**) butane only, and (**C**) mixed feed.

The total paraffin conversion and selectivity were high for mixed feed, because the reactants act as a diluent to each other's shifting equilibrium towards the product side [38]. The presence of Sn (II, IV) in the $NaBH_4$ reduced catalyst improves the catalytic activity and increases the conversion and selectivity [33].

### 2.2.4. TGA Analysis of Coke Deposition on Spent Catalysts

The amount of coke deposited during the dehydrogenation reaction was measured using thermogravimetric (TG) analysis from room temperature to 700 °C. The detailed experimental methodology is provided in Section 3.4. The amount of coke removed under oxygen atmosphere with respect to temperature is shown in Figure 8a for the $NaBH_4$ reduced catalyst and Figure 8b for the hydrogen reduced catalyst. The temperature programmed oxidation profile (TPO) based on the coke removal rate is presented in Figure 8c for the $NaBH_4$ reduced catalyst and Figure 8d for the hydrogen reduced catalyst. It was reported that the complete combustion of coke will happen under 400 °C in air atmosphere [1]. From Figure 8a,b, it can be noted that the complete combustion of coke deposited on both catalysts happened below 400 °C.

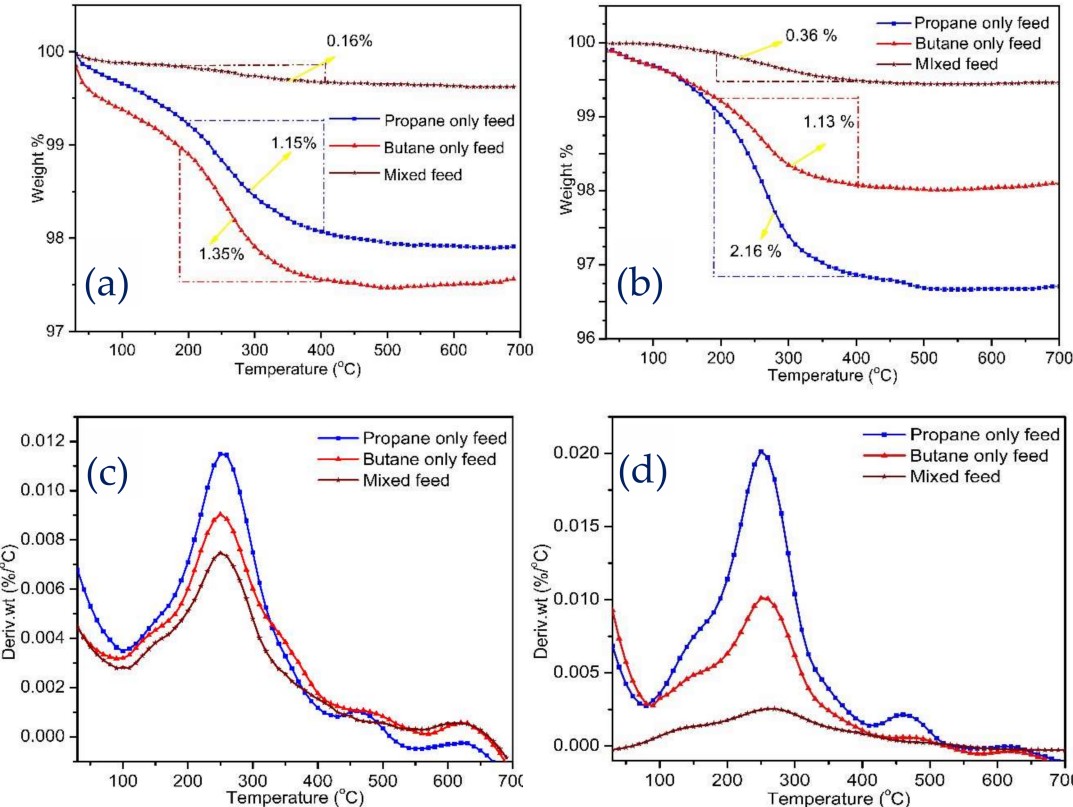

**Figure 8.** Thermogravimetric profiles of deactivated catalysts under oxygen atmosphere: (**a**) NaBH$_4$ reduced catalyst and (**b**) H$_2$ reduced catalyst, Differential thermal analyses of deactivated catalysts under oxygen atmosphere (**c**) NaBH$_4$ reduced catalyst and (**d**) H$_2$ reduced catalyst

Only one peak was observed during the differential thermal analysis of both spent catalysts, as shown in Figure 8c,d, between 200 °C and 320 °C. The exothermic peak between 200 °C and 320 °C represents the carbon deposition on the metal surface, which is the result of the catalyzing effect of metal towards oxidation of coke deposited on it [13,25,34]. From the DTA curves for both catalysts, it can be inferred that the coke is deposited only on the metal particles.

Table 4 shows the amount of coke deposited during dehydrogenation after 240 min. The weight percentage of coke deposited on the NaBH$_4$ reduced catalyst was 1.15%, 1.35%, and 0.91% for propane only, butane only, and mixed feed, respectively. Similarly, the amount of coke deposited on the hydrogen reduced catalyst was found to be 2.16%, 1.13%, and 0.36% for propane, butane, and mixed feed, respectively. It was interesting to note that the percentage of coke deposited was much lower during mixed-paraffin feed when compared with individual paraffins for both catalysts. The amount of coke deposited on the NaBH$_4$ reduced catalyst for individual feed is almost same. Whereas the amount of coke deposited during propane only feed is higher when compared with butene only for the hydrogen reduced catalyst.

**Table 4.** Amount of coke deposited on catalysts during paraffin dehydrogenation.

| Catalytic Deactivation Properties | NaBH$_4$ Reduced Catalyst | | | Hydrogen Reduced Catalyst | | |
|---|---|---|---|---|---|---|
| | Propane Feed Only | Butane Feed Only | Mixed Feed | Propane Feed Only | Butane Feed Only | Mixed Feed |
| Amount of coke% [a] | 1.15 | 1.35 | 0.16 | 2.16 | 1.13 | 0.36 |

[a] From thermogravimetric analysis.

From Figure 8c,d, it can also be seen that the mass loss derivative for the NaBH$_4$ reduced catalyst is much lower when compared with that of the hydrogen reduced catalyst. As mentioned earlier, this is because of the difference in the catalyzing effect of metal alloys present on both catalysts.

The higher amount of coke deposited for individual paraffin feed on the hydrogen reduced catalyst is the result of the presence of large PtSn alloy particles on the surface, resulting in faster deactivation of the catalyst. The lower amount of coke deposited for mixed-paraffin feed is the result of lower partial pressure of hydrocarbons because of the mutual dilution effect. This is also the reason for higher conversion under mixed-paraffin feed when compared with individual paraffin feed. The amount of coke deposited from the current studies is much lower when compared with other similar studies under dehydrogenation conditions. This might be because the feed has a mixture of two paraffins, which does not allow the deposited coke to hydrogenate, and thus results in a higher amount of coke deposition. Similar results were obtained from propane dehydrogenation co-fed with hydrogen [39].

From the results discussed in catalyst characterization (Section 2.1), individual paraffin dehydrogenation (Section 2.2.2), mixed-paraffin feed dehydrogenation (Section 2.2.3), and TG analysis (Section 2.2.4), the NaBH$_4$ reduced catalyst has higher stability when compared with the hydrogen reduced catalyst, and mixed-paraffin feed dehydrogenation results in higher conversion and low coking, without affecting the olefin selectivity. It is recommended that mixed-paraffin feed dehydrogenation be used along with the NaBH$_4$ reduced catalyst.

## 3. Materials and Methods

### 3.1. Catalyst Preparation

All the chemicals used were of analytical grade and purchased from Sigma Aldrich (St. Louis, MO, USA). The bimetallic Pt–Sn/$\alpha$-Al$_2$O$_3$ catalysts were prepared by the wetness impregnation method using H$_2$PtCl$_6$·6H$_2$O and SnCl$_2$ as precursors. Double distilled (2D water) was used to prepare solutions. Then, 1 mL of HCl acidified Sn precursor solution was mixed with 1 mL of Pt precursor to get a red wine color, which indicates the formation of Pt–Sn alloy. The precursor solution was added drop by drop on Al$_2$O$_3$ support, for uniform distribution of the catalyst precursors on the surface by wetness impregnation. The Pt–Sn impregnated Al$_2$O$_3$ was then dried at 120 °C for 12 h, followed by calcination at 550 °C for 6 h in a muffle furnace. The calcined bimetallic Pt–Sn/Al$_2$O$_3$ was dispersed in 100 mL of distilled water under continuous stirring. The catalyst slurry was reduced with 30% excess of NaBH$_4$ using a syringe pump at a rate of 60 µl per min, and washed with 2D water until neutral pH. Finally, the reduced catalyst slurry was dried for 12 h at 120 °C using a hot air oven in air atmosphere. The prepared catalyst was named to the NaBH$_4$ reduced catalyst. A similar procedure was followed until the calcination for the hydrogen reduced catalyst, the calcined catalyst was reduced with gas phase hydrogen at 100 sccm for 4 h at 600 °C, instead of reducing with NaBH$_4$. This catalyst was named the H$_2$ reduced catalyst. For both catalysts, the nominal amounts of Pt and Sn were fixed to be 0.5 wt% and 0.7 wt%, respectively.

### 3.2. Characterization

The X-ray diffraction spectra of the prepared Pt–Sn/Al$_2$O$_3$ catalyst were obtained using Ultima IV XRD Rigaku Corporation, Tokyo, Japan, using Cu K$\alpha$ radiation source with a $\lambda$ range of 0.15418 nm.

The samples were scanned in 2θ angle of 10° to 90° with a scanning rate of 5°/min. The surface area, pore volume, and average pore diameter were measured using nitrogen sorption in Quantachrome Nova 1200e instrument. The samples were degassed at 300 °C for 4 h under vacuum condition. X-ray photon spectroscopy spectra were obtained on AXIS ULTRA DLD (Kraton analytical) with 0.6 eV resolution using Al Kα radiation of 1486.6 eV. The Pt 4f and Sn 3d spectra were de-convoluted using CasaXPS software. The scanning electron microscope and energy dispersive spectroscopy images were captured using JOEL-6393, Tokyo, Japan. The product gases from the reactor were analyzed using GCMS-QP2010 Ultra (Shimadzu, Kyoto, Japan), with Rt-Alumina BOND/Na$_2$SO$_4$ (Restek Corporation, Bellefonte, PA, USA) and flame-ionization detector (FID). The thermal analysis of the spent catalysts was obtained using DST Q600 (TA Instruments, New Castle, DE, USA).

### 3.3. Dehydrogenation of Individual and Mixed Paraffin Feed

The process and instrumentation diagram (P&ID) of the experimental setup, developed for mixed paraffin dehydrogenation, is given in Figure 9. The experimental set up consists of a fixed bed reactor, gas mass flow controllers, and temperature and pressure measurement devices. The fixed bed reactor is fabricated from Inconel 600. The inner diameter and length of the Inconel reactor are 1 cm and 45 cm, respectively. One gram of catalyst was packed between two stationary quartz wool supports. An inline fine wire (1.6 mm diameter) K-type thermocouple from Omega Engineering was used to measure the catalyst bed temperature. Temperatures and pressures at both inlet and outlet of the reactors were measured using inline thermocouples and pressure gauges. The desired reactor temperature was maintained using an electrically heated tubular furnace. The reactor pressure was maintained using a backpressure regulator. The gas flow rates of all the gases were controlled using mass flow controllers. The reaction was conducted at 600 °C and 10 psi pressure with 11 to 15 h$^{-1}$ weight hourly space velocity. The weight hourly space velocity (*WHSV*) was calculated using Equation (7) for all experiments that were conducted.

$$WHSV = \frac{Gas\,mass\,flow\,rate}{Mass\,of\,the\,catalyst} \tag{7}$$

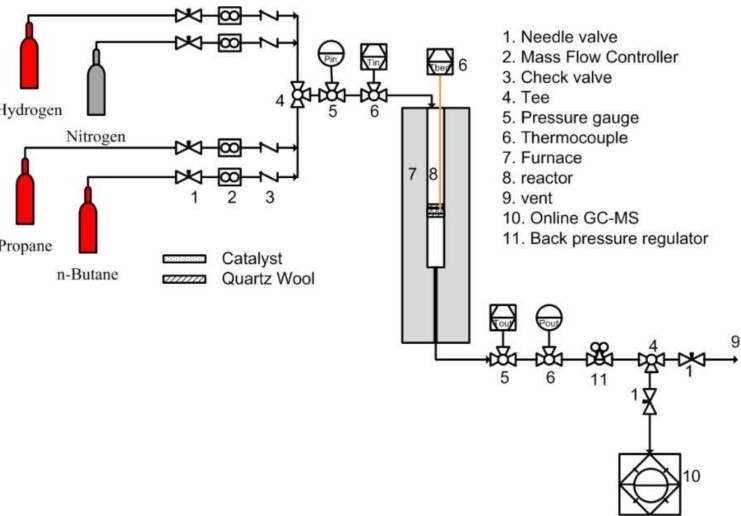

**Figure 9.** Process and instrumentation diagram (P&ID) of the experimental setup for mixed-paraffin dehydrogenation studies.

The paraffin (butane and propane) conversion and the olefin selectivity were calculated using the formulae given below.

$$Propane\, Conversion\left(X_{C_3H_8}\right) = \frac{\left(n_{C_3H_8}\right)_{in} - \left(n_{C_3H_8}\right)_{out}}{\left(n_{C_3H_8}\right)_{in}} \times 100, \tag{8}$$

$$Butane\, Conversion\left(X_{C_4H_{10}}\right) = \frac{\left(n_{C_4H_{10}}\right)_{in} - \left(n_{C_4H_{10}}\right)_{out}}{\left(n_{C_4H_{10}}\right)_{in}} \times 100, \tag{9}$$

$$Total\, Olefin\, Selectivity = \frac{\left(n_{C_3H_6} + n_{C_4H_8}\right)_{out}}{\left(\sum_i n_i\right)_{out} - \left[\left(n_{C_3H_6} + n_{C_4H_8}\right)_{out} + \left(n_{C_3H_8} + n_{C_4H_{10}}\right)_{out}\right]} \times 100 \tag{10}$$

where

$\left(n_{C_3H_8}\right)_{in}$ is number of moles of propane at the inlet,

$\left(n_{C_3H_8}\right)_{out}$ is number of moles of propane at the outlet,

$\left(n_{C_4H_{10}}\right)_{in}$ is number of moles of butane at the inlet,

$\left(n_{C_4H_{10}}\right)_{out}$ is number of moles of butane at the outlet,

$\left(n_{C_3H_6}\right)_{out}$ is number of moles of propylene at the inlet,

$\left(n_{C_4H_8}\right)_{out}$ is number of moles of butylene at the outlet,

$\left(\sum_i n_i\right)_{out}$ is number moles of all hydrocarbons at the outlet.

As selectivity is generally defined as the ratio of the desired product to the undesired product formed, the selectivity value may exceed 100% [40]. Ragula et al. obtained olefin selectivity as high as 300% through modeling of mixed-paraffin dehydrogenation [41]. It is to be noted that, for the calculation of olefin selectivity using Equation (10), only relevant terms that are applicable are used.

### 3.4. TG-DTA Studies of Spent Catalysts under Oxygen Atmosphere

The coke deposited on $NaBH_4$ and hydrogen reduced catalysts was studied using thermogravimetry and differential thermal analysis under air atmosphere. The air used was zero-air, which contains only oxygen and nitrogen and is free from hydrocarbons. The TG-DTA experiments were carried out from room temperature to 700 °C, with approximately 3 mg of spent catalyst at a heating rate of 20 °C/min under 100 sccm air flow in a thermogravimetric balance that combines heat flux type differential thermal analysis (DTA) and thermogravimetric analysis (TGA), SDT-Q6000, supplied by TA Instruments, New Castle, DE, USA. The temperature in the instrument is maintained using an electrically heated furnace. The sample weight as function of temperature against an inert standard was measured using thermogravimetric balance and the temperature of both samples and inert standard was measured using a thermocouple.

### 4. Conclusions

The effect of the reducing agent on the synthesis of the Pt–Sn/α-Al$_2$O$_3$ catalyst and its performance on paraffin conversion and olefin selectivity during individual and mixed-paraffin feed dehydrogenation was investigated in this work. The objective behind the addition of Sn to Pt was to improve the stability of the catalyst. Pt–Sn/α-Al$_2$O$_3$ catalysts were prepared using $NaBH_4$ and hydrogen as the reducing agents. The prepared catalysts were characterized using BET, SEM-EDS, XRD, and XPS. Both Pt$_3$Sn and PtSn alloy phases were identified from the characterization results. A higher amount of the Pt$_3$Sn phase was observed when compared with the PtSn phase in both

catalysts. The PtSn alloys were finely dispersed over the support in the NaBH$_4$ reduced catalyst when compared with the hydrogen reduced catalyst. Sn(II,IV) and Sn(0) were observed in the NaBH$_4$ reduced catalyst, whereas only Sn(0) was observed in the hydrogen reduced catalyst. Sn(0) is a poison for the dehydrogenation reaction. The dehydrogenation study showed that the NaBH$_4$ reduced catalyst has a good catalytic performance when compared with the hydrogen reduced catalyst, owing to the favorable formation of PtSn intermetallic states. On the basis of the physicochemical properties of the catalyst and catalytic performance and stability during dehydrogenation, NaBH$_4$ is a potential alternative as a reducing agent for the synthesis of the Pt–Sn/$\alpha$-Al$_2$O$_3$ catalyst.

Individual paraffin dehydrogenation was compared with mixed paraffin dehydrogenation, and the results showed higher paraffin conversion during mixed paraffin dehydrogenation when compared with individual paraffin dehydrogenation, without affecting the selectivity to olefins. Therefore, mixed-paraffin feed dehydrogenation has advantages for olefin production. This results in reduced plants' capital and operational costs. The TG-DTA studies showed the amount of coke deposited was lower during mixed-paraffin feed when compared with individual paraffin feed; this will lead to additional savings on the regeneration of catalyst. The lower amount of coke deposited is the result of the variation in secondary reactions and the mutual dilution effect during mixed-paraffin feed dehydrogenation, which is evident from the product profiles from individual and mixed-paraffin feed, which shifts the equilibrium towards favorable products. A the NaBH$_4$ reduced catalyst has higher stability when compared with the hydrogen reduced catalyst and mixed-paraffin feed dehydrogenation results in higher conversion and low coking without affecting the olefin selectivity, it is recommended that mixed-paraffin feed dehydrogenation be used along with the NaBH$_4$ reduced catalyst.

**Supplementary Materials:** The following are available online at http://www.mdpi.com/2073-4344/10/1/113/s1, Figure S1: Fresh feed profiles obtained from GC-MS analysis: (A) propane only, (B) butane only, and (C) mixed feed; Table S1: GC-MS analysis result of mixed feed.

**Author Contributions:** Concept Development—U.B.R.R.; Experimental setup development—S.A.K. and U.B.R.R.; Experimental Studies—S.A.K., Results Analysis—S.A.K. and U.B.R.R.; Original Draft Preparation—S.A.K.; Review and editing—S.A.K. and U.B.R.R. All authors have read and agreed to the published version of the manuscript.

**Funding:** This work was financially supported by Science and Engineering Research Board (SERB), India, for development of Experimental setup (File No: YSS/2014/000510).

**Acknowledgments:** Suresh A.K. also acknowledges Council of Scientific & Industrial Research (CSIR), India, for his fellowship support (File No: 09/942(0017)2K18 EMR-I). Authors acknowledge Ministry of Human Research and Development (MHRD), India for allowing the use of GC-MS and TGA facilities (Ref. No.5-6/2013 TS.VII (General) dated 8th May 2013, Department of Science and Technology (DST)-FIST, India for the use of the FE-SEM and EDS facility (Ref. No. SR/FST/ETI-416/2016), dated 16th December 2016, for catalyst characterization and Dr. T. Senthilkumar, Scientist-MDP, Central Institute for Research on Cotton Technology, Mumbai, India for helping with BET Analysis.

**Conflicts of Interest:** The authors have no conflict of interest.

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
