# Peer review of "Effect of Reduction of Pt–Sn/α-Al2O3 on Catalytic Dehydrogenation of Mixed-Paraffin Feed"

_catalysts, doi:10.3390/catal10010113_

Round 1
Reviewer 1 Report
Review of the manuscript “Effect of Reduction of Pt-Sn/α-Al2O3 on Catalytic Dehydrogenation of Mixed-paraffin Feed” (catalysts-675744) by Suresh A.K. and Udaya Bhaskar Reddy Ragula
The Authors described the effect of Pt-Sn/α-Al2O3 catalyst reduction method on dehydrogenation of mixed-light paraffins to olefins. The bimetallic Pt-Sn/α-Al2O3 catalysts were prepared by two different methods: liquid phase reduction with NaBH4; and gas phase reduction with hydrogen. The catalysts were examined using X-Ray Diffraction (XRD), Scanning Electron Microscopy (SEM), Energy Dispersive X-ray Spectroscopy (EDS), X-ray Photoelectron Spectroscopy (XPS), Thermo-Gravimetric Analysis (TGA), and Brunauer–Emmett–Teller (BET) method.
Below, I am listing a number of points that may help improving the quality of the manuscript:
In introduction, on page 3, style of citation is wrong. Please correct all as follow: Zangeneh et al. [20] , etc. Page 3, line 89, Please correct: Zhang et al. [20], similarly, line 92 – should be Zhang et al. [24] Page 3, line 94, Should be “Mostyly cases hydrogen was used as reducing agent at temperature higher than 550oC [19]. If it is "Mostly" one citation is not enough. An intensive correction of the English language is necessary. Some expression do not have subject, predicate, etc. Where is Figure. 22a, 22b, etc.? What is the expremiental error in case of BET method? In my opinion SBET should not contain the number after a decimal point. All the used abbreviations should be explained. Table 1. Please express the total pore volume in non exponential form. Additionally, use common unit, e. g. cm3/g.My overall impression after reading the manuscript. The whole manuscript is rather messy. Tables, figure captions are not in accordance with the guidelines of the Instruction for authors (https://www.mdpi.com/journal/catalysts/instructions). The text is underdeveloped and requires thorough changes and refinement in detail. I am really sorry, The article is full of errors, which makes it impossible to review it further.
The reviewer's task is not to guess what the author meant and correct errors, but to read and write reviews.
In the present form, the manuscript is neither suitable for reviewing nor for publishing.
Taking into account large number of experiment carried out, the manuscript could be consider for publication if a intensive rewriting of the text, especially of discussion was done.
Author Response
Dear Reviewer-1,
Thank you for your time and effort that was spent in providing constructive feed back on the submitted manuscript titled "Effect of Reduction of Pt-Sn/α-Al2O3 on Catalytic Dehydrogenation of Mixed-paraffin Feed” (catalysts-675744) by Suresh A.K. and Udaya Bhaskar Reddy Ragula.
We now wish to submit the correction made as per your comments.
Please let us know if further improvements / modifications are required in the manuscript.
Thank you,
Suresk A.K. and Udaya Bhaskar Reddy Ragula

Reviewer 2 Report
This manuscript very interesting because they give information about the study of NaBH4 was the simple and promising alternative reduction method for Pt-Sn/Al2O3 catalyst synthesis in favor of paraffin dehydrogenation. And this studies also explain about mixed feed dehydrogenation can give higher conversions without affecting the selectivity to olefins significantly. However, we found some missing information in your manuscript following by: General comment 1. Please checking the English grammar. 2. Please check the word, still many typo and double font in your paper. 3. Please check the similarity of your paper and improve it. 4. Abstract in this paper need more information about specific result and objective in your research. 5. Please improve the conclusion. Make it clear and can explain all point in author manuscript. Specific comment 1. Your introduction section was informative. However, you should give more information about objective of your research and give the explanation about your method compare the reference method you use. 2. In BET measurement section, line 115. Need improvement. • Please improve it and give citation if your data based on reference. • Please give more information and explanation about BET measurement in your paper. 3. In XRD measurement section, figure. 2a and 2b. In this section the author explains the result about the characteristic sample and compare it with reference result. However • That part need more information and explanation more deeply about the result characteristic. • And why in figure b at shows the higher position in 39.0 degree? • The author just explain that your result was matching with reference result, however you should give explanation about that phenomena and effect to the catalyst. 4. In XRD measurement section. Your explanation about XPS analysis was not complete. You didn’t explain clearly about your result graphic. Please improve it and give the informative explanation in your result. 5. In Blank test section, the method you describe in your paper not clear. When you use the blank test in your experiment? and how the effect your method with the experiment test? Please improve it and make the explanation clearly and informative. 6. In individual paraffin dehydrogenation section. Please improve your sentence and writing procedures. So messy. 7. In line 287, you explain about thermogravimetric analysis experiment with heating rate 20C from room temperature 700C. Why you use that temperature? And how you measure that? Please give more information about that temperature and how you manage that temperature in your experiment. 8. In dehydrogenation of individual and mixed paraffin feed section, the method you describe in your paper need more information. We difficult to understand about your explanation. Please improve it and make the explanation step by step.

Author Response
Dear Reviewer-2,
Thank you for your time and effort that was spent in providing constructive feed back on the submitted manuscript titled "Effect of Reduction of Pt-Sn/α-Al2O3 on Catalytic Dehydrogenation of Mixed-paraffin Feed” (catalysts-675744) by Suresh A.K. and Udaya Bhaskar Reddy Ragula.
We now wish to submit the correction made as per your comments.
Please let us know if further improvements / modifications are required in the manuscript.
Thank you,
Suresk A.K. and Udaya Bhaskar Reddy Ragula

Reviewer 3 Report
This work aims to study the dehydrogenation of mixed-paraffin feeds. Particularly the research study focuses on the effect of reduction of Pt-Sn/alfa-alumina catalyst. In addition they compare individual and mixed-paraffin dehydrogenation.
I have some comments prior to the publication of this article:
INTRODUCTION SECTION
It provides enough background about the state of the art of this topic, and includes prior updated works. However, more effort could be made to manifest the advantages of using NaBH4 as reducing agent and the improvements and benefits found in the literature regarding catalyst activity and deactivation.
MATERIALS AND METHODS
Please describe the methodology used for TGA analysis: combustion temperature, heating ramp, etc.
I recommend to move Figure 9 to Supporting Information.
I suggest to add capital letters to whsv (WHSV). I also recommend to improve the quality of presentation of equations 8 and 9 (without table) and proper nomenclature for reactants and products (Fi, Fo, etc.)
I wonder if this is the common way of the defining selectivity for dehydrogenation processes. If not, I recommend to define it in order not to exceed 100%.
RESULTS
I consider that the authors should make greater effort to emphasize more the benefits of NaBH4 reduction. I recommend to correlate catalystic results with catalyst properties to better understand the performance observed.
More explanation is required to understand the benefits of mixed paraffin dehydrogenation: synergies, etc.
Please move TGA analysis information to the experimental section.
I recoomend to include TPO profiles rather than TGA profiles to see if different cokes are observed and to identify better the evolution and development degree of each coke (combustion T, etc.)
Regards
Author Response
Dear Reviewer-3,
Thank you for your time and effort that was spent in providing constructive feed back on the submitted manuscript titled "Effect of Reduction of Pt-Sn/α-Al2O3 on Catalytic Dehydrogenation of Mixed-paraffin Feed” (catalysts-675744) by Suresh A.K. and Udaya Bhaskar Reddy Ragula.
We now wish to submit the correction made as per your comments.
Please let us know if further improvements / modifications are required in the manuscript.
Thank you,
Suresk A.K. and Udaya Bhaskar Reddy Ragula

Round 2
Reviewer 2 Report
This manuscript very interesting because they give information about the study of NaBH4 was the simple and promising alternative reduction method for Pt-Sn/Al2O3 catalyst synthesis in favor of paraffin dehydrogenation. And this studies also explain about mixed feed dehydrogenation can give higher conversions without affecting the selectivity to olefins significantly. However, we found some missing information in your manuscript following by:
General comment
The explanation was improving from the author previous manuscript. We can easily understand what research you doing in this manuscript. However, the author should check the figure quality in your manuscript, especially for figure 1, 2 and 8.Reviewer 3 Report
The authors have taken into account the suggestions of the reviewers.